# Wearable and Portable GPS Solutions for Monitoring Mobility in Dementia: A Systematic Review

**DOI:** 10.3390/s22093336

**Published:** 2022-04-27

**Authors:** Anisha Cullen, Md Khadimul Anam Mazhar, Matthew D. Smith, Fiona E. Lithander, Mícheál Ó Breasail, Emily J. Henderson

**Affiliations:** 1Population Health Sciences, Bristol Medical School, University of Bristol, 1–5 Whiteladies Road, Bristol BS8 1NU, UK; qc20383@alumni.bristol.ac.uk (M.K.A.M.); matthew.smith@bristol.ac.uk (M.D.S.); fiona.lithander@bristol.ac.uk (F.E.L.); micheal.obreasail@bristol.ac.uk (M.Ó.B.); emily.henderson@bristol.ac.uk (E.J.H.); 2Liggins Institute, University of Auckland, 85 Park Road, Grafton, Auckland 1142, New Zealand; 3Older Peoples Unit, NHS Foundation Trust, Royal United Hospital, Bath BN1 3NG, UK

**Keywords:** Alzheimer’s disease, remote monitoring, sensors, GPS, movement/mobility, wearable technology

## Abstract

Dementia is the most common neurodegenerative disorder globally. Disease progression is marked by declining cognitive function accompanied by changes in mobility. Increased sedentary behaviour and, conversely, wandering and becoming lost are common. Global positioning system (GPS) solutions are increasingly used by caregivers to locate missing people with dementia (PwD) but also offer a non-invasive means of monitoring mobility patterns in PwD. We performed a systematic search across five databases to identify papers published since 2000, where wearable or portable GPS was used to monitor mobility in patients with common dementias or mild cognitive impairment (MCI). Disease and GPS-specific vocabulary were searched singly, and then in combination, identifying 3004 papers. Following deduplication, we screened 1972 papers and retained 17 studies after a full-text review. Only 1/17 studies used a wrist-worn GPS solution, while all others were variously located on the patient. We characterised the studies using a conceptual framework, finding marked heterogeneity in the number and complexity of reported GPS-derived mobility outcomes. *Duration* was the most frequently reported category of mobility reported (15/17), followed by *out of home* (14/17), and *stop* and *trajectory* (both 10/17). Future research would benefit from greater standardisation and harmonisation of reporting which would enable GPS-derived measures of mobility to be incorporated more robustly into clinical trials.

## 1. Introduction

### 1.1. Dementia

Dementia is a neurodegenerative disorder with growing global prevalence. There are currently over 850,000 people living with dementia in the UK [1], with an estimated 55 million cases globally [2]. Demographic changes within high income countries, and increases in longevity in low and middle-income countries, will lead to a projected 78 million people living with dementia in 2030 and 139 million by 2050 [2]. A 2010 paper estimating the cost of disorders of the brain across 30 European countries estimated the direct and indirect costs of dementia to be €105 billion (Purchasing Power Parity 2010) [3]. In 2019, it was estimated that the total global societal cost of dementia stood at $1.3 trillion, with costs expected to exceed $2.8 trillion by 2030 [2]. As such, there is a pressing economic and societal need for technological solutions to improve care for, and reduce the care costs of, people with dementia (PwD). 

Dementia encompasses a spectrum of disorders which are broadly characterised by an individual’s progressive loss of cognitive ability. The most common type is Alzheimer’s disease, which is associated with the accumulation of amyloid plaques and cellular neurofibrillary tangles [4]. Other dementias vary in their aetiology, onset and specific symptom profile with sub-types including vascular dementias, dementia with Lewy bodies and dementias in disorders such as Huntington’s and Parkinson’s disease. 

Dementias can also have an adverse effect on an individual’s ability to move. This effect is caused by multiple factors, such as the loss of volition and motivation, a reduction in the need to walk to tasks that can no longer be cognitively performed, low mood, and a direct effect on higher-order gait control [5]. Over time, sedentary behaviour may predominate due to decreased activity and movement [6], leading to social isolation, spatial disorientation, and depressive symptoms. As disease severity increases, wandering behaviours may manifest [7], imparting a risk of personal injury (e.g., a road traffic accident) and environmental exposure (e.g., hypothermia). Cumulatively, these factors may negatively impact a PwD’s cognition, cardiovascular health, brain plasticity and mood. These changes have a profound impact on family members, who often act as informal caregivers (CGs), with the European Social Survey estimating in 2014 that family carers provide three quarters of the total long-term care to dependent older adults [8]. 

### 1.2. GPS-Derived Mobility Data as a Potential Tool in Clinical Decision Making

Mobility refers to the physical movement of individuals where movement is defined temporally, spatially and with complimentary descriptors, such as mode of transportation [9]. Therefore, mobility reflects an individual’s ability to navigate and interact with their environment [10]. Within the remit of mobility, life-space mobility (LSM) is a distinct concept, defined as the spatial extent in which an individual moves within a specified period [11]. Life-space area intertwines with LSM, describing the spatial area that an individual commonly navigates [12]. As mobility is intrinsic to the disease trajectory of dementia [13], longitudinal measurements across the disease course capture a dynamic state. Changes in this metric may, therefore, herald a need for greater support to maintain independence, fitness, and wellbeing. 

Currently, the Life Space Assessment (LSA), a questionnaire-based tool, is the most widely validated means to quantify LSM in older adults [14,15,16], though other tools have been used [15,17]. Travel diaries are commonly used to capture other important aspects of mobility not measured by the LSA, such as mode of transport or type of activity [18]. However, these techniques are retrospective, and, hence, are subject to recall bias. Furthermore, cognitive impairment limits the reliability of these retrospective measures. 

Technology-based monitoring solutions, such as global navigation satellite systems (GNSS), most commonly the global positioning system (GPS), may facilitate the passive assessment of mobility patterns. Efficiently acquiring satellite position is essential in capturing accurate GPS data. Therefore, GPS cannot detect mobility patterns accurately in indoor environments. GPS data consists of spatial data, using latitude and longitude coordinates to report location, and temporal data. Recent validation studies in healthy older adults have suggested that GPS-derived spatial mobility indicators show reasonable agreement with the used LSA tool [19,20,21]. GPS devices can accurately and precisely determine a user’s position to within an accuracy of metres; such devices have the potential to capture sufficient GPS data from which a vast range of mobility indicators can be derived. In turn, this has the potential to report mobility outcomes beyond that of conventional questionnaire-based tools and to detect fluctuations in behaviour that a PwD might not detect themselves [22]. In turn, these could act as potential proxies of disease progression and severity, consequently informing the health and social care of these patients. 

### 1.3. Use of GPS in Patients with Dementia 

GPS has become ubiquitous in portable consumer devices, such as smartphones [23] and navigation systems [24], as well as wearable devices, such as smartwatches [25]. As defined by Ye et al., a wearable device is ’a product controlled by electronic components and software that can be incorporated into clothing or worn on the body like accessories‘ [26]. Preliminary work based on such devices has captured data from which mobility outcomes have been derived; most commonly *the average time spent out of home* and *the average number of locations* visited by the user [27]. With a wide range of GPS-derived mobility outcomes coupled with sophisticated statistical or machine learning (ML) approaches, there is the potential to differentiate mild-to-moderate Alzheimer disease patients from people without Alzheimer’s disease [28]. This offers an important opportunity to incorporate measures of mobility into healthcare, with particular benefit to PwD and CGs. 

### 1.4. Processing Tools and Algorithms Used for GPS Data Analysis 

Due to the large volumes of GPS data collected, extensive data processing is essential to produce GPS-derived mobility indicators that reflect the full range of an individual’s mobility [19]. There are two main approaches to GPS data processing: (1) Post-processing, where the GPS data are processed after the study period has been completed, and (2) Real-time processing where the GPS data are processed while the individual is wearing the device [29]. Post-processing is more commonly used, however, within dementia research; real-time processing is often used to alert a CG when a PwD leaves a ‘safe zone’ [30]. In post-processing, ML is often implemented to predict an individual’s patterns of mobility. ML methods commonly include neural networks [31], density-based clustering algorithms [22] and support-vector machines [32], using software such as MatLab [28] and V-Analytics [33] for processing. Currently there is no gold standard methodology to process GPS data to derive mobility indicators. 

### 1.5. Assessing the Breadth of Mobility Measures Applied in the Dementia Literature

A recent conceptual framework developed by Fillekes and colleagues [34] provides an important tool for synthesizing GPS-derived mobility indicators that can be used in health and ageing research. Additionally, it offers a framework to critique and compare the breadth of mobility outcomes reported in existing studies [34]. This framework is divided into the domains, *space, time, movement scope* and *attribute* with each domain containing categories. The *space* domain contains categories detailing the spatial distribution of the GPS data (LSM) while the other domains capture additional aspects of mobility not associated with LSM. *Time* focuses on the temporal distribution of the data, such as the time spent out of the home. *Movement scope* details if the GPS data captures the individual’s trajectory or solely their stop locations. Finally, *attribute* includes categories that do not fall in the previously described domains. As part of this framework, Fillekes et al. have suggested GPS-derived mobility indicators that capture all the categories across the domains [34]. 

### 1.6. Contribution of This Paper to the Literature 

The increase in commercial and research-orientated GPS devices offers the opportunity to assess the mobility of PwD. However, despite the increasing availability and usage of GPS technology, there has been little attempt to synthesis this literature. Therefore, in this review, we aim to systematically summarise the current breadth of the available biomedical literature on GPS, in particular wearable GPS, as a solution to passively monitor mobility in PwD. Our objectives are:To describe the use of GPS in dementia care and management;To identify the extent to which wearable solutions have been used;To assess the quality of the studies identified using the conceptual Fillekes framework.

## 2. Materials and Methods

### 2.1. Contribution of This Paper to the Literature 

A systematic search using MEDLINE, EMBASE, AMED (Allied and Complementary Medicine), APA PsycInfo and IEEE Xplore was conducted from, and including, the year 2000 up until 2 February 2022. Searches were restricted to human studies and performed in alignment with the Preferred Reporting Items for Systematic Review and Meta-Analysis (PRISMA) guidelines [35]. Search terms included a combination of technical keywords surrounding GPS devices, such as “Geographic Information System”, “wearables” and “GPS”. These terms were searched in isolation and then combined with the disease search terms such as “Alzheimer”, “Dementia”, “Huntington *” and “Mild Cognitive Impairment” (see Appendix A). These search terms were developed in accordance with the NICE guidelines [36]. PICO (population, intervention, comparison, outcome) principles were used: (P) PwD or mild cognitive impairment (MCI); (I) wearable devices containing GPS; (C) none; (O) GPS-derived mobility indicators. 

### 2.2. Eligibility

Inclusion criteria were: (a) GPS sensor technology primarily used to assess mobility; (b) the GPS technology was a wearable or portable device [26]; (c) the technology was implemented on a population with the most common types of dementia (Alzheimer’s disease, vascular dementias, Lewy body dementia, and Huntington’s disease) or MCI. Studies were excluded where: (a) they did not report location or mobility-based outcomes; (b) they did not undergo peer-review; (c) they were a conference abstract or a systematic review; (d) the primary focus of the study was on GPS technology as a navigation aid or alerting a caregiver if the PwD left a specific zone (i.e., geofencing). 

### 2.3. Study Selection

References were uploaded to EndNote reference management software to remove duplicates. The references were then imported to Rayyan for review. A.C. conducted the title and abstract review alongside M.Ó.B., and, where there were disagreements on the inclusion of a study, M.S. determined the final decision. From screening of the titles and abstracts using the eligibility criteria, 1944 references were excluded and 28 included for full-text review. A total of 185 studies were excluded, as they were identified as a publication type outside the criteria (i.e., conference abstracts or systematic review), or they did not undergo peer-review. After assessment of the full text conducted by A.C., M.Ó.B. and F.E.L., 17 studies were determined to have met the eligibility criteria and underwent data extraction. A summary of the screening process is shown in Figure 1. 

### 2.4. Data Extraction and Study Synthesis 

Data extraction from the full text of the included studies was completed by A.C. where the following were identified: author (year), study population, device (s) and location carried on the body, duration, sampling frequency, study design, GPS-derived outcomes, data processing details and key findings. The included studies were assessed on the number and range of GPS-derived mobility outcomes using the Fillekes framework. Within this framework, mobility indicators were classified based on the domains: *space*, *time*, *movement scope* and *attributes* (see Section 1.4). The categories encompassed by each of these domains were (*space*: *count*, *extent*, and *shape/distribution*), (*time*: *duration*, *timing* and, *temporal distribution*), (*movement scope*: *stop*, *move* and *trajectory*) and (*attribute*: *out of home*, *transport mode* and *further attribute*) [34]. 

## 3. Results

### 3.1. Study Characteristics

In total, 3004 papers were found after the initial search; once duplicates were removed, 1972 studies remained (Figure 1). An overview of the 17 included studies is provided in Table 1. Five of the 17 studies focused on the mobility of a cognitively impaired population (either PwD or MCI), while 12/17 studies compared the mobility of a cognitively impaired population with an age-matched healthy control group. Two studies included CGs in their population to determine if the GPS device improved CG quality of life. Nine studies were part of a single project (Senior tracking, SenTra) [27,33,37,38,39,40,41,42,43]. 

A single study by Chung et al. employed a truly wearable GPS solution (Garmin™ Vivoactive HR smartwatch) worn on the PwD’s wrist [48]. Thorpe et al. also used a smartwatch but in combination with a smartphone which provided the GPS logging [22]. The nine studies from the SenTra project used a wristwatch with a radio frequency (RF) transmitter which communicated with a portable GPS receiver to verify if the individual was carrying the GPS device. All other studies used portable devices which could be worn on the person using additional accessories, such as a pouch, or were carried in pockets, bags, or used a standalone smartphone. 

The GPS monitoring duration varied across the studies with four weeks used in ten studies, including the nine studies involved in the SenTra project. The minimum duration was 3–5 days in the Tung et al. study [28] and the maximum duration was eight weeks, used by Thorpe et al. and Bayat et al. [22,46]. However, in the Bayat et al. study only four weeks of the collected data were processed to compare with the information from the LSA tool which assessed mobility over four weeks. The study by Thorpe et al. however, processed all the data collected across the eight week duration. Sampling frequency varied across the studies from a maximum of 1 Hz to 0.017 Hz [22,46]. Commonly, 0.2 Hz and 0.1 Hz were used, including in the SenTra project studies. One study (Thorpe et al.) reported an intermittent sampling frequency ranging from 1000 Hz to 0.003 Hz [22].

### 3.2. GPS Data Processing

Across the included studies, a range of processing methods were implemented to derive GPS mobility outcomes. In the studies involved in the SenTra project, wear-time of the GPS receiver, determined from the additional radio frequency (RF) wristwatch and home RF monitoring system, was used to assess the validity of the data. Kaspar et al. defined a valid day as having <1 h of missing data whereas Oswald et al. considered a day valid if there were no invalid hours, with an invalid hour having <30 min of valid GPS data [37,38]. The home radius, the set radius around the home coordinates used to determine if the participant left their home, varied across studies with values ranging from approximately 25 m [28] to 500 m [49]. Studies commonly agreed that a visited location is a GPS data point which has the same coordinates for at least 5 min.

A variety of data packages were used to process the data including MatLab, V Analytics and R. The methodology for processing the GPS data was rarely detailed in the included studies. The Bayat et al. studies detailed their ML methodologies, namely the density-based spatial clustering of applications with noise algorithm, and the Trajectory segmentation method, which were used in combination to extract the location of the participant [47]. Several studies also used the convex hull algorithm to capture the convex hull area, the smallest convex shape that contains all the GPS recorded coordinates, which allowed the life space area to be captured [22,28,48]. In four of the studies a geographic information system (GIS) was used in alignment with the GPS data to retrieve information about the nature of the location that the participants visited, for example, a supermarket, leisure centre or medical centre [45,46,50,51].

### 3.3. GPS-Derived Mobility Indicators 

Applying the Fillekes framework to the studies described in Table 1 revealed that the Bayat et al. study provided the most comprehensive set of mobility indicators, considering all but one of the categories, *shape distribution,* as shown in Table 2 [51]. Due to the positive associations with mobility and the categories in the *movement scope* domain, all of the identified studies included at least one of these categories. Either *stop* or *trajectory* were derived in ten studies and *move* recorded in nine studies. Across the *space* and *time* domains, the most frequently derived categories were *count* (nine studies) and *duration* (fifteen studies). However, across these domains the categories that focused on qualitative data were less well covered and no studies reported *shape/distribution*; two studies reported *timing* and two studies reported *temporal distribution.* As most of the identified studies focused on the time the participants spent out of home, the category *out of home* was frequently used (fourteen studies). Additionally, seven studies included the category *further attribute*, and seven studies reported additionally derived outcomes, including walking speed (specifically those within the SenTra project) and the type of activity carried out by the participant.

## 4. Discussion

We will discuss the overall findings of our review, thereafter, presenting the broader context in which these findings can be interpreted in the wider role of GPS in the care, treatment, and management of PwD. Additionally, we consider future research and the potential considerations associated with such research. 

### 4.1. Overall Findings

GPS technologies are emerging as a solution to monitor mobility and life space patterns in PwD. Across nine of the included studies [22,37,38,40,42,45,48,49,50], GPS-derived mobility outcomes broadly agreed with travel diaries completed by participants. This is congruent with work in healthy older adults and populations with non-dementia neurodegenerative disorders [19,20,21]. Of the included studies, 9/17 were part of a single project (SenTra) and made use of three devices: an RF wristwatch, a GPS receiver and a home monitoring unit [44]. The GPS receiver had a short battery life of only 12 h when set to a sampling rate of 0.1 Hz. All devices contained an RF component to determine if the individual was carrying the GPS receiver outside the house determining GPS wear-time. Oswald et al. reported that, on average, participants carried the GPS receiver out of the home for 88% of the total captured hours [38]. Additionally, the SenTra tracking kit had a short battery life of only 12 h when set to a sampling rate of 0.1 Hz. Only one study (Chung et al.) used a GPS-enabled smartwatch, with high PwD and CG compliance reported in the observed dyad [48]. On one hand the PwD found the smartwatch easy to use and convenient. However, the number of buttons on the device rendered it difficult to use and on occasion the user struggled to turn it on and inadvertently switched it off. Additionally, the smartwatch had a relatively short battery life (13 h in GPS tracking mode [52]). The remainder of the GPS solutions were commercially available GPS devices which the participant carried in their pocket, bag or purse. Examples include, QStarz BT-1000XT, a multipurpose GPS device, and the SafeTracks Prime Mobile Device which is a dedicated PwD tracker. Although these devices are not as compact as a wristwatch, they have a significantly longer battery life, with QStarz BT-1000XT having a 42 h life at its lowest sampling rate frequency (0.001 Hz) [53]. 

### 4.2. Mobility Behaviour of PwD 

Several of the included studies compared the mobility behaviour of PwD with age matched HCs and those with MCI. Findings showed that PwD were more mobility restricted [42] and travelled a shorter distance from home during the day compared with HCs [33]. These findings suggest PwD experience an increase in sedentary behaviour [6], as well as a reduction in the need to walk to perform tasks that can no longer be cognitively sequenced [5]. Differences in the mobility of participants with MCI were also noted, with them spending less time out of their home and visiting fewer locations compared with HCs. Although the reduction in mobility was not as high in participants with MCI with PwD, it highlights that cognitive impairment, regardless of severity, can cause a reduction in an individual’s mobility behaviour. 

### 4.3. GPS-Derived Outcomes and Processing Methods 

Within the Fillekes framework, the majority of included studies reported quantitative GPS-derived outcomes, such as the number of locations visited (*count* and *stop*), the time spent out of the home (*duration* and *out of home*) and the number of tracks (*move*). Time spent out of home was calculated by determining when an individual was outside of the home radius, defined as a set external radius around the home coordinates, and for how long. When *count* was reported in a study the *stop* category (GPS coordinates in the same location for >5 min) was also derived. Within 3/9 studies involved in the SenTra project, *move* was reported and defined as any movement less than 5 km/h. *Trajectory* (which combines *stop* and *move*) was reported in 10/17 studies and was visualised using packages such as Google Earth and ArcMap to chart an individual’s overall movement. These basic algorithms highlight the relative ease of deriving such categories. Bayat et al. adapted and expanded on the basic methodology of determining *stop* locations by differentiating *stop* locations into a *full signal stop* (i.e., a set of continuous GPS coordinates at the same location) or a *no signal stop* (i.e., where the GPS signal is lost/ no motion detected) [47]. Using these definitions, the cluster methodology mapped the *stop* locations of the individuals considering if the GPS signal was lost or motion was undetected. 

Few studies derived the more qualitative GPS outcomes of the *attributes* domain, such as activity type (*further attribute)* or *transport mode*. Only Bayat et al. reported their methodology to determine activity type [46]. Key ML entropy methods (random, heterogeneous spatial, spatiotemporal) were used to determine the randomness of an individual’s mobility behaviour and thus to predict their future activities based on known information about their daily pattern of visited locations. Spatiotemporal entropy patterns predicted a 5% chance, on average, that a PwD would choose a location in a random manner compared with an 8% chance in HCs [46].

Within the wider literature, a combination of the rule-based machine learning method and predefined heuristic rules are used to determine activity type [47]. Bohte and Matt [54] used this methodology, combining location-specific information with the last known *stop* location of the individual; if their last known *stop* location was a 50 m radius from a known location it could be assumed they were visiting that location [54]. Other studies have used probabilistic methods [55], though only at an aggregated level (not an individual level). As previously highlighted, the entropy methods used by Bayat et al. have the potential to predict the destination of PwD [46]. However, at present, there is no gold standard methodology to capture activity type. Instead, studies commonly use a combination of GPS-derived data with information from a travel diary, which are subject to recall bias and are not suitable for a cognitively impaired population. Future research needs to develop suitable methodologies to determine qualitative data outcomes or to validate entropy methods more widely. 

### 4.4. The Extent to Which Wearable Solutions Have Been Used 

A single study used a truly wearable GPS device in a wristwatch form [48]. However, the majority of studies used commercially available GPS devices that could be carried on the participants in a pocket, bag or purse. It remains debatable if such solutions are wearable as they are not incorporated into clothing or worn on the body [26]. Conversely, if a device is within a pouch/pocket attached to the individual’s body it may in some respects be considered wearable. Some studies (including the SenTra project) specified that participants could carry devices in a pouch worn around the abdomen, but they did not indicate if such accessories were provided or were used by participants [44]. Therefore, it can be assumed across the studies that they relied on the participants remembering to carry the GPS device when they left the house. Consequently, the most common reasons for missing GPS data were either due to the participant forgetting to carry the device or the device running out of battery. However, the commercially available portable GPS devices, such as the QStarz BT-1000XT used by Sturge et al. [50], have a long battery life, up to 42 h under certain configurations, meaning, if charged and used appropriately, they can capture a complete 24 h period [53]. As it is unlikely that a cognitively impaired population will all remember to correctly charge their devices throughout the study duration, it is arguably inevitable for devices to run out of battery in some cases. Therefore, ensuring the device is wearable will aid in reducing the amount of missing data. 

While many novel devices have been developed that surmount some of the challenges surrounding battery life and wearability [56,57], they remain in the prototype stage and, to date, none have been widely trialled in a clinical population. 

### 4.5. Device Use and Acceptability in PwD 

All of the included studies focused on quantitative data, but some also included qualitative data (*n* = 2). Chung et al., a dyadic study, considered the device acceptance by the PwD, and CG impressions on whether it reduced overall burden on both PwD and CG [48]. Both were accepting of the form of the device, though the PwD sometimes struggled with unintentionally turning off the device. The PwD and CG were accepting of the impact of GPS-based monitoring on privacy, if the information could be of use to healthcare professionals. Crucially, significant CG support was needed to remind the PwD to wear the device and to ensure that tracking was active. As such, this type of device may be ineffective in a study population of PwDs without CG support. The Thorpe et al. study solely consisted of PwD using a smartphone for GPS tracking; 4/6 of the participants who already used smartphones thought the device easy to use and were accepting of it [22]. The other participants were not accepting of the smartphone; one found the rubber strap, provided to make the smartphone wearable, uncomfortable and the other found the device too big and heavy. Overall, participants found the device beneficial, and all but one of the participants wished to continue using the device after the study duration. In contrast, the GPS tracking kit in the SenTra project had an extensive set up which was reported as making high cognitive and behavioural demands on participants [38]. This suggests PwD are more accepting of devices that are simple, compact and easy to use as they can incorporate this into their daily lives without too much additional cognitive burden. 

Although the lack of qualitative data across the included studies hampers our ability to determine the most appropriate GPS device to monitor mobility in PwD, the wider literature provides useful insights into GPS device selection. Megges et al. reported the acceptability of two different models of GPS watch in 17 patient–CG dyads with devices rated on usability, telephone function, overall design features, font, buttons, and battery life [58]. For both products, usability, determined by the International Standardization Organisation Norm (ISONORM) 9241/10 usability scale [59], ranged from fair to good. However, product satisfaction was significantly lower at home for both products [58]. Freiesleben et al. explored barriers to the adoption of GPS technologies for dementia care through interdisciplinary stakeholder meetings consisting of professionals working in business (*n* = 7), healthcare (*n* = 6), and research (*n* = 9) [60]. Øderud et al. explored the day-to-day administration of GPS devices in a cohort of PwD, focusing on who was responsible for device charging, ensuring it was switched on/off, and, where applicable, who assisted the PwD during their daily activities [61]. The study found that health care professionals, usually nursing home based, administered devices for approximately 50% of PwD, family CGs administered for about 30% of PwD, while in only 4% of cases, the PwD administered the device themselves. Sustained device use was low with 46% and 12% of participants using the device beyond one and two years, respectively [61]. The most common reason for dropout cited was the PwD no longer being able to perform outdoor activities. 

### 4.6. Strengths and Limitations 

This review searched the biomedical and technological literature, with studies only included that tested GPS technology on PwD in real-world conditions. While numerous novel approaches to applying GPS and similar technologies were detailed in the technological literature, few of these had been tested beyond the prototype phase and many had only been tested in healthy controls and not PwD. We limited our search to studies conducted from 2000 onwards as initial scoping of the literature suggested few studies in this area before 2010, though we cannot discount the possibility of earlier studies. We made a pragmatic decision to focus on common dementia sub-types and, whilst we used broad search criteria, we recognize that we may have excluded studies of rarer dementia subtypes, such as HIV-related dementia and others. Due to the marked heterogeneity of the studies, we pragmatically decided against performing a formal risk-of-bias assessment but acknowledge that many of the included studies are at high risk of bias due to modest sample sizes and the lack of control groups. 

## 5. Conclusions

This review summarises the current use of GPS devices in dementia research to assess the mobility of PwD and highlights the heterogeneity of GPS-derived mobility outcomes reported across existing studies. At present, few clinical studies have incorporated the full breadth of GPS mobility domains and categories that can be derived from raw GPS data. Therefore, future research would benefit from greater standardisation and harmonisation of reporting to ensure GPS-derived mobility outcomes are incorporated more robustly into clinical trials. In turn, this would enable the relationship between mobility and dementia disease severity or progression to be explored further.

In the interim, we advocate that studies aim to capture the maximum number of mobility categories described in the Fillekes framework to develop a deeper understanding of an individual’s mobility. Studies should particularly focus on the categories detailing LSM and those most relevant to clinical practice, disability, function and wellbeing.

Currently, there is no ideal GPS tracking device for clinical research. Such a device would benefit from being developed alongside PwD and CGs to ensure it captures meaningful data and is acceptable to the end user (s). A compact, wearable device with a simple interface and long battery life would enable higher quality data collection and enhance user experience. 

## Figures and Tables

**Figure 1 sensors-22-03336-f001:**
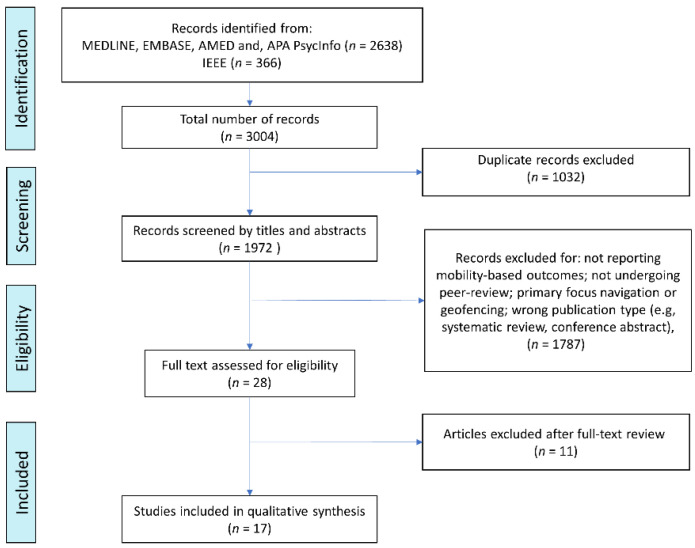
PRISMA flowchart summarising the search process and results.

**Table 1 sensors-22-03336-t001:** Summary of identified papers where GPS has been used in patients with dementia disorders. Participant age is presented as mean (SD) where available.

Author (Year)	Study Population	Device(s) and Location Carried	Duration	Sampling Frequency	Study Design	GPS-Derived Outcomes	Data Processing Details	Key Findings
Oswald et al. [38] (2010)	PwD *n* = 6, MCI *n* = 6, HC *n* = 7 (Participants between age 63–80).	SenTra device: GPS receiver, a RF transmitter wristwatch, and a home RF monitoring system [44]. Location: GPS device located in pouch or bag [44].	4 weeks	0.2 Hz	Observational, cross-sectional study	Distance travelled, Walking speed, Distance from home, Daily mobility activity	The GPS data were transmitted via the GPRS protocol to a project server. A valid hour was ≥30 min of valid GPS data; a day was valid only if there were no invalid hours. Full time analysis was carried out on valid days (methodology of processing was not stated).	This study established that the future proposed SenTra project was feasible. However, the SenTra tracking kit placed high cognitive and behavioural demands on participants.
Shoval et al. [33] (2011)	PwD *n* = 7 (mean age 81.9), MCI *n* = 21 (mean age 78.3), HC *n* = 13 (mean age 72.9).	SenTra kit [44]	4 weeks	0.1 Hz	Observational, cross-sectional study	Distance from home, Time OOH	GPS data transmitted via the GPRS protocol to a project server. Using a combination of a GIS and the recorded locations of the participant, the distance from home was calculated. This information was visualized on a ‘spider-web diagram.’	Participants with cognitive impairment travelled shorter distances from home during the day compared with HCs. PwD had a smaller spatial range compared to those with MCI.
Werner et al. [27] (2012)	PwD *n* = 16, MCI *n* = 34, HC *n* = 26, CG *n* = 66 (Participants aged 63 or older).	SenTra kit [44]	4 weeks	0.1 Hz	Observational, cross-sectional study	Time spent OOH per day, Time spent walking per day, Number of visited nodes, Number of walking tracks per day, Average walking distance, Average walking speed	GPS data transmitted via the GPRS protocol to a project server. From GPS data a *node* was defined as a stopping point lasting >5 min. A *track* was the pathway between nodes. Detail was not presented on the processing methods to gather GPS derived outcomes.	The greater the mobility of PwD (with mobility defined through the GPS derived outcomes), the less burden placed on CGs.
Wahl et al. [39] (2013)	MCI *n =* 76 (mean age 72.9), HC *n* = 146 (mean age 72.5).	SenTra kit [44]	4 weeks	0.1 Hz	Observational, cross-sectional study	Time spent OOH per day, Number of visited locations	A valid day was when <1 h of missing data was observed. A visited location was defined as a GPS coordinate staying in the same location for >5 min.	The mean number of visited locations was higher in HCs than those with MCI.
Tung et al. [28] (2014)	PwD *n* = 19 (mean age 70.7), HC *n* = 33 (mean age 73.7).	GPS receiver on smartphone chipset (Qualcomm RTR6285, Qualcomm Inc., San Diego, CA, USA) Location: Pocket	3–5 days	1 Hz	Observational, cross-sectional study	Life-space area, Distance from home, Time OOH	GPS coordinates were projected to a 2D plane using Matlab R12. Home radius was set to 25 m around home coordinates determined from the participants address and Google Earth. A convex hull, calculated using the standard convex hull operation, was used to determine the area and perimeter measures. The Euclidean distance from the home coordinates was calculated and a distance time series was produced to determine the time spent OOH and distance from home were calculated.	Reduced mobility was observed in PwD compared to HCs, using measurements of the area and perimeter of the convex hull.
Wettstein et al. [40] (2014a)	PwD *n* = 35 (mean age 74.1), MCI *n* = 76 (mean age 72.9), HC *n* = 146 (mean age 72.5)	SenTra kit [44]	4 weeks	0.2 Hz	Observational, cross-sectional study	Walking distance, Walking speed, Walking duration, Time spent OOH, Number of places visited, Number of walking tracks per day	GPS data transmitted via the GPRS protocol to a project server. A valid day had to have OOH behaviour and <1 h of missing GPS data. A visited location was defined as GPS coordinates in the same location for >5 min. A walking track was considered as movement less than 5 km/h.	In PwD, higher walking distance and walking speed were positively correlated with environmental mastery (how capable an individual feels with using environmental resources).
Wettstein et al. [41] (2014b)	As per Wettstein et al. [40]	SenTra kit [44]	4 weeks	0.2 Hz	Observational, cross-sectional study	Time spent OOH, Number of places visited	The same data processing method was used as per Wettstein et al. [40]. However, the walking tracks were not processed as this was not a GPS derived outcome for this study.	Behavioural competence was significantly lower in PwD than both MCI and HC. The mean number of activities carried out was also lower in PwD compared with MCI and HCs.
Kaspar et al. [37] (2015)	PwD *n* = 16, MCI *n* = 30, HC *n =* 95 (Participants were in the age range 50–84).	SenTra kit [44]	4 weeks	0.2 Hz	Case Control study	Time spent OOH, Average walking distance, Type of activity, Type of transport	The GPS data were transmitted via the GPRS protocol to a project server. A valid day had <1 h of missing data. Spatial GPS data was interpreted using complex algorithms (specific type not stated), which integrated compound measures, such as acceleration and velocity, alongside geographical background data to distinguish transport modes.	The authors were unable to establish a strong relationship between daily mood and an individual’s mobility.
Wettstein et al. [42] (2015a)	As per Wettstein et al. [40]	SenTra kit [44]	4 weeks	0.2 Hz	Observational, cross-sectional study	Walking distance, Walking speed, Walking duration, Time spent OOH, Number of places visited, Number of walking tracks per day	The same data processing method was used as Wettstein et al. [40]. An addition of this study was the cluster method which used GPS-derived outcomes to identify whether the participants were ‘mobility restricted’, ‘outdoor oriented’ or ‘walkers’.	The mobility patterns in older people were heterogenous. However, it was identified that there was a higher proportion of cognitively impaired individuals in the cluster defined as having restricted mobility.
Wettstein et al. [43] (2015b)	As per Wettstein et al. [40]	SenTra kit [44]	4 weeks	0.2 Hz	Observational, cross-sectional study	Walking distance, Walking speed, Walking duration, Time spent OOH, Number of places visited, Number of walking tracks per day	Data processing method the same as Wettstein et al. [40]	The three cognitive ability groups did not significantly differ in OOH walking indicators (e.g., walking speed). However, OOH mobility indicators (time OOH, number of visited locations) were lowest in PwD.
Harada et al. [45] (2019)	PwD *n* = 147 (The mean age of the *n =* 192 baseline participants was 76.3 but the age was not stated for those included in the final study)	Globalsat DG-200 Data Logger Location: Pocket	2 weeks	0.033 Hz	Secondary analysis of a randomised controlled trial	Time spent OOH per day	GPS data was processed in accordance with the GIS system (ArcGIS for Desktop 10.3: Esri Japan Incorporation: Tokyo, Japan). Home radius set to 100 m around the home coordinates; the time spent OOH was determined using this radius. Validity of a day was defined as wear ≥10 h, location started and ended in the home area, no poor connection during the time OOH and, the participant stated they wore the device in their travel diary.	In PwD, a stronger social network was positively correlated with greater time spent OOH. However, no relationship between environmental factors and time spent OOH was observed in PwD.
Thorpe et al. [22] (2019)	PwD *n* = 6 (Mean age 69.7)	Smartphone (Nexus 5) and a Smartwatch (Sony Smartwatch 3) Location: Pocket (smartphone) and Wrist (smartwatch).	8 weeks	Ranging from 1000 Hz to 0.003 Hz	Longitudinal study	MCP, Action range, Total distance covered outside the home, Time spent OOH, Time spent moving between locations, Number of places visited, Number of trips	The GPS data was filtered in alignment with the upper limit set at 25 m accuracy. The *stop* and *moves* were determined from the trajectory with the DBSCAN method applied to determine locations. A stop was defined as GPS coordinates in the same location for >5 min. The MCP was calculated using the R function to determine the smallest convex polygon around the data points. The action range was the geodesic distance from home coordinates and the GPS data [23].	Digital monitoring of mobility and activity has the potential to detect fluctuations in behaviour that the participant might not detect themselves.
Bayat et al. [46] (2021)	PwD *n* = 7, HC *n* = 8 (All participants were ≥65 or older).	SafeTracks Prime Mobile GPS Device Location: Pocket	8 weeks	0.017 Hz	Case control study	Number of destinations, Sequence of destinations, Time spent at each destination	4 of the 8 weeks of captured GPS data were extracted. Home location of each participant was determined using DBSCAN algorithm. The trajectory segmentation method [47] extracted the locations visited by each participant. Extracted destinations were clustered and each destination was assigned a cluster ID [47]. Different entropy methods (random, heterogeneous spatial and spatiotemporal) and algorithms were used to assess randomness of individuals mobility.	There was lower spatial and temporal randomness in mobility patterns in PwD compared to HC. Therefore, across the collected data there was a 5% chance, on average, that a PwD would choose a location at random but an 8% chance in HC.
Chung et al. [48] (2021)	PwD *n* = 1, CG *n* = 1 (PwD 64, CG 62)	Garmin™ Vivoactive HR Location: wrist watch	1 week	Not stated	Case study	Total distance moved, Movement speed, Convex hull area, Total wear time, Location (home vs. other), Total time OOH, Total time at home, Heart rate	The GPS data were extracted in TCX and CSV formats. The participant wore the device longer than the intended 7-day study period therefore generating 9 days of complete GPS data. GPS track plots used to describe locations visited with total distance moved and speed of movement determined for each track. LSM visualized by plotting and calculating the convex hull of GPS points using mapview package (CITE). Home radius was set as ≤1000 ft around home coordinates.	The participant engaged in OOH activities every day from late morning until the evening. The travel diary correlated with the GPS-derived outcomes and provided additional information on the type of activity the participant carried out.
Liddle et al. [49] (2021)	PwD *n* = 3, MCI *n* = 15 (Participants mean age 86.7)	Smartphone based GPS system Location: Pocket	Required 105 to 240 h of GPS data.	Not stated	Longitudinal observational study	Life space area, Time at home, Maximum distance from home, Trips OOH, Time left at home	Custom algorithms (not stated) were used to create metrics. The locations extracted from the GPS data were plotted to visualize the life space area and the shape and perimeter of the life space area were analysed. The home area was defined as 500 m from the home location and the time spent OOH was when the participant left the home radius and did not return for a period > 5 min.	The authors found no relationship between life space and cognition. However, an association with life space and driving status was found with non-drivers having a lower life space compared with drivers.
Sturge et al. [50] (2021)	PwD *n* = 2, MCI *n* = 5 (Participants were aged 59–93).	QStarz BT—1000X	2 weeks	Not stated	Observational, cross-sectional study	Visited locations, Distance from home, Life Space Area	GPS data extracted and processed in Microsoft Excel then imported into V-Analytics to store the participants locations and trips over the study period and for time-space movement analysis. Activities were created if GPS location points were connected within an 80 m radius for >5 min. GPS locations exceeding this radius were considered as a distinct trip. Activities were imported into ArcMap 10.5.1. to visualize participants’ spatial movement with activities then defined into routine activity space (<7.5 km of the home coordinates) and occasional activity space (>7.5 km).	Cognitively impaired individuals still engaged in activities beyond their neighbourhood area.
Bayat et al. [51] (2022)	PwD *n* = 7, HC *n* = 8 (All participants were ≥ 65).	SafeTracks Prime Mobile GPS device Location: Pocket, purse or bag	4 weeks	Not stated	Case control study	Maximum distance from home, Radius of gyration, Life space area, Number of destinations, Number of unique destinations, Time at home, Time OOH, Time on foot, Time in vehicle, Trip time period, Total number of trips, Outdoor activity duration, Types of activities	Data processing as described by Bayat et al. was used in this study [46]. A distance-based probabilistic model based on Google places, API, was used to retrieve information about visited locations of the participants to define their OOH activities (i.e., shopping, leisure, medical services).	PwD undertook more medical-related and fewer sport-related activities compared to HCs. PwD spent less time walking than cognitively intact individuals.

GPS = global positioning system, GIS = geographic information system, GPRS = general packet radio service, RF = radio frequency, GSM = global system for mobile communications, DBSCAN = density-based spatial clustering of applications with noise, PwD = people with dementia, CG = caregivers, MCI = mild cognitive impairment, HC = healthy control, LSM = life space mobility, OOH = out of home, API = application programme interface.

**Table 2 sensors-22-03336-t002:** Characterisation of dementia studies based on the GPS-derived mobility indicators used, according to the characteristic aspects of the conceptual framework by Fillekes et al. [34], where mobility indicators are classified based on their analytical and characteristic aspects, which are then grouped into further thematically organized categories.ju.

Study	Space			Time			Movement Scope			Attribute			Total Number of Outcomes
	Count	Extent	Shape/Distribution	Duration	Timing	Temporal Distribution	Stop	Move	Trajectory	Out of Home	Transport Mode	Further Attribute	
Oswald et al. (2010) [38]								●	●			●	**3**
Shoval et al. (2011) [33]		●		●	●				●	●			**5**
Werner et al. (2012) [27]	●			●			●	●		●		●	**6**
Wahl et al. (2013) [39]	●			●			●			●			**4**
Tung et al. (2014) [28]		●		●					●	●			**4**
Wettstein et al. (2014a) [40]	●			●			●	●		●		●	**6**
Wettstein et al. (2014b) [41]	●			●			●			●			**4**
Kaspar et al. (2015) [37]		●		●				●	●	●	●	●	**7**
Wettstein et al. (2015a) [42]	●			●			●	●		●		●	**6**
Wettstein et al. (2015b) [43]	●			●			●	●		●		●	**6**
Harada et al. (2019) [45]				●					●	●			**3**
Thorpe et al. (2019) [22]	●	●		●			●	●	●	●			**7**
Bayat et al. (2021) [46]	●			●		●	●						**4**
Chung et al. (2021) [48]		●		●					●	●			**4**
Liddle et al. (2021) [49]		●		●					●	●			**4**
Sturge et al. (2021) [50]		●					●	●	●				**4**
Bayat et al. (2022) [51]	●	●		●	●	●	●	●	●	●	●	●	**11**
**Total number of studies per category**	**9**	**8**	**0**	**15**	**2**	**2**	**10**	**9**	**10**	**14**	**2**	**7**	

## Data Availability

Not applicable.

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
