# Peer review of "Wearable and Portable GPS Solutions for Monitoring Mobility in Dementia: A Systematic Review"

_sensors, 2022, doi:10.3390/s22093336_

Round 1

Reviewer 1 Report

The present work entitled "Wearable and Portable GPS Solutions for Monitoring Mobility in Dementia: A Systematic Review" is a sistematic review and for this reason it could be satisfy specific criteria.

Introduction section is complete and presented an updated biography. Just one observation, i think that at line 51: "Huntingdon" could be Huntington.

Authors proposed two really good written paragraph (1.2 and 1.3) about GPS solutions and in particular about GPS applications in dementia.

Paragraph 1.6 is really useful to help readers in understanding the aims of the study and how it is postioning in the current literature.

The study includes inclusion and esclusion criteria and Prisma guideline but i can't find risk of bias.

I can't find the "Risk of bias", it's necessary to write a review to analyze the risk of bias, could authors add a section about this?

Discussion and conclusion sections reflect results and well describe all new aspects of the work.

Reviewer 2 Report

Dear authors,

please find my comments in the attached.

Others,  authors complain about non-existing solutions e.g.

-for wearing check/sensor

-for long battery life

-attachment to body, to not forget device.

There is a publication of year 2014 providing already a solution to these problems through a back-plaster. I should be cited for completeness:

https://ieeexplore.ieee.org/document/6798920

Andre SchwarzmeierJürgen BesserRobert WeigelGeorg FischerDietmar Kissinger  A compact back-plaster sensor node for dementia and Alzheimer patient care, 2014 IEEE Sensors Applications Symposium (SAS)

Overall I like the ambition to identify and standardize common metrics for quantifying mobility. They are indeed lacking. The authors provide a convincing argumentation why they are needed and based on an exhaustive literature search they also make reasonable proposals. Overall I like the paper very much.

Thank you for interesting work.

Reviewer
